# CHAIN-OF-THOUGHT PREDICTIVE CONTROL

## ABSTRACT

We study generalizable policy learning from demonstrations for complex low-level control tasks (e.g., contact-rich object manipulations). We propose a novel hierarchical imitation learning method that utilizes scalable, albeit sub-optimal, demonstrations. Firstly, we propose an observation space-agnostic approach that efficiently discovers the multi-step subgoal decomposition (sequences of key observations) of the demos in an unsupervised manner. By grouping temporarily close and functionally similar actions into subskill-level segments, the discovered breakpoints (the segment boundaries) constitute a chain of planning steps (i.e., the chain-of-thought) to complete the task. Next, we propose a Transformer-based design that effectively learns to predict the chain-of-thought (CoT) as the high-level guidance for low-level action. We couple action and CoT predictions via prompt tokens and a hybrid masking strategy, which enable dynamically updated CoT guidance at test time and improve feature representation of the trajectory for generalizable policy learning. Our method, named Chain-of-Thought Predictive Control (CoTPC), consistently surpasses existing strong baselines on a wide range of challenging low-level manipulation tasks with scalable yet sub-optimal demos.

## 1 INTRODUCTION

Hierarchical RL (HRL) [31] has attracted much attention in the AI community as a promising direction for sample-efficient and generalizable policy learning. HRL tackles complex sequential decision-making problems by decomposing them into simpler and smaller sub-problems via temporal abstractions (the so-called chain-of-thought [82]). In addition, many adopt a two-stage policy and possess the planning capabilities for high-level actions (i.e., subgoals or options) to achieve generalizability. On the other hand, imitation learning (IL) remains one of the most powerful approaches to training autonomous agents. Without densely labelled rewards or on-policy / online interactions, IL usually casts policy learning as (self-)supervised learning with the potential to leverage large-scale pre-collected demonstrations, usually with Transformer, as inspired by the recent success of large language models (LLMs). An obstacle in building foundational decision-making models [87] remains the better use of sub-optimal demonstrations. In this paper, we study hierarchical IL from sub-optimal demonstrations for low-level control tasks.

Despite the recent progress [8, 20, 71, 47, 2], it remains extremely challenging to solve low-level control tasks such as contact-rich object manipulations by IL in a scalable manner. Usually, the demonstrations are inherently sub-optimal because of the underlying contact dynamics [59] and the way they are produced. The undesirable properties, such as being non-Markovian, noisy, discontinuous, and random, pose great challenges in both the optimization and the generalization of the imitators (see detailed discussion in Appendix A). We find that, by adopting the hierarchical principles (i.e., temporal abstraction and high-level planning) into our Transformer-based design, we can enjoy large-scale (albeit sub-optimal) demonstrations for their performance boost on solving challenging tasks. To achieve this, we first propose an unsupervised chain-of-thought discovery strategy to generate CoT supervision from the demonstrations. We then design our model to learn to dynamically generate CoT guidance for better low-level action predictions.

Specifically, we consider the multi-step subgoal decomposition of a task into a chain of planning steps as its chain-of-thought (inspired by CoT [82] and PC [86]). As part one of our contribution, we propose an observation space-agnostic approach that efficiently discovers the chain-of-thought (CoT), defined as a sequence of key observations, of the demos in an unsupervised manner. We propose to group temporarily close and functionally similar actions into subskill-level segments. Then the

breakpoints (the segment boundaries) naturally constitute the CoT that represents the high-level task completion process. For part two, we propose a novel Transformer-based design that effectively learns to predict the CoT jointly with the low-level actions. This coupled prediction mechanism is achieved by adding additional prompt tokens at the beginning of the context history and by adopting a hybrid masking strategy. As a result, CoT guidance is dynamically updated at each step and better feature representation of the trajectories is learned, eventually improving the generalizability of the low-level action prediction process.

We call our method Chain-of-Thought Predictive Control (CoTPC). From an optimization perspective, it learns faster from sub-optimal demos by utilizing the subgoals (CoTs) that are usually more robust and admit less variance. From a generalization perspective, it uses Transformers [5] to improve generalization with CoT planning, which is learned from the unsupervisedly discovered CoTs from the demos. We evaluate CoTPC on several challenging low-level control tasks (Moving Maze, Franka-Kitch and ManiSkill2) and verify its design with ablation studies. We find that CoTPC consistently outperforms several strong baselines.

## 2 RELATED WORK

**Learning from Demonstrations (LfD)**    Learning interactive agents from pre-collected demos has been popular due to its effectiveness and scalability. Roughly speaking, there are three categories: offline RL, online RL with auxiliary demos, and behavior cloning (BC). While offline RL approaches [40, 22, 44, 41, 39, 8, 81] usually require demonstration with densely labelled rewards and the methods that augment online RL with demos [27, 37, 69, 53, 66, 28, 58, 74] rely on on-policy interactions, BC [60] formulates fully supervised or self-supervised learning problems with better practicality and is adopted widely, especially in robotics [88, 20, 62, 90, 4, 64, 21, 89]. However, a well-known shortcoming of BC is the compounding error [69], usually caused by the distribution shift between the demo and the test-time trajectories. Various methods were proposed to tackle it [68, 69, 75, 43, 78, 3, 7]. Other issues include non-Markovity [50], discontinuity [20], randomness and noisiness [70, 83] of the demos that results in great compounding errors of neural policies during inference (see Appendix A for detailed discussions).

**LfD as Sequence Modeling**    A recent trend in offline policy learning is to relax the Markovian assumption of policies, partially due to the widespread success of sequence models [23, 12, 79] where model expressiveness and capacity are preferred over algorithmic sophistication. Among these, [15, 49] study one-shot imitation learning, [48, 74] explore behavior priors from demos, [8, 47, 34, 71, 2, 33] examine different modeling strategies for policy learning. In particular, methods based on Transformers [79, 5] are extremely popular due to their simplicity and effectiveness.

**Hierarchical Approaches in Sequence Modeling and RL**    Chain-of-Thought [82] refers to the general strategy of solving multi-step problems by decomposing them into a sequence of intermediate steps. It has recently been applied extensively in a variety of problems such as mathematical reasoning [46, 14], program execution [67, 54], commonsense or general reasoning [65, 13, 45, 82], and robotics [84, 91, 36, 24, 86, 72, 32]. Similar ideas in the context of HRL can date back to Feudal RL [16] and the option framework [76]. Inspired by these approaches, ours focuses on the imitation learning setup (without reward labels or online interactions) for low-level control tasks. Note that while Procedure cloning [86] shares a similar name to our paper, it suffers from certain limitations that make it much less applicable (see detailed discussion in Appendix).

**Demonstrations for Robotics Tasks**    In practice, the optimality assumption of the demos is usually violated for robotics tasks. Demos involving low-level actions primarily come in three forms: human demo captured via teleoperation [42, 80], expert demo generated by RL agents [51, 52, 9, 35], or those found by planners (e.g., heuristics, sampling, search) [25, 63, 18]. These demos are in general sub-optimal due to either human bias, imperfect RL agents, or the nature of the planners. In this paper, we mostly focus on learning from demos generated by planners in ManiSkill2 [25], a benchmark not currently saturated for IL while being adequately challenging (see details in Sec. 5.3).

## 3 PRELIMINARIES

**MDP Formulation**  One of the most common ways to formulate a sequential decision-making problem is via a Markov Decision Process, or MDP [30], defined as a 6-tuple $\langle S, A, \mathcal{T}, \mathcal{R}, \rho_0, \gamma \rangle$, with a state space $S$, an action space $A$, a Markovian transition probability $\mathcal{T} : S \times A \to \Delta(S)$, a reward function $\mathcal{R} : S \times A \to \mathbb{R}$, an initial state distribution $\rho_0$, and a discount factor $\gamma \in [0, 1]$. An agent interacts with the environment characterized by $\mathcal{T}$ and $\mathcal{R}$ according to a policy $\pi : S \to \Delta(A)$. We denote a trajectory as $\tau_\pi$ as a sequence of $(s_0, a_0, s_1, a_0, ..., s_t, a_t)$ by taking actions according to a policy $\pi$. At each time step, the agent receives a reward signal $r_t \sim \mathcal{R}(s_t, a_t)$. The distribution of trajectories induced by $\pi$ is denoted as $P(\tau_\pi)$. The goal is to find the optimal policy $\pi^*$ that maximizes the expected return $\mathbb{E}_{\tau \sim \pi}[\sum_t \gamma^t r_t]$. Notice that, in robotics tasks and many real-world applications, the reward is at the best only sparsely given (e.g., a binary success signal) or given only after the trajectory ends (non-Markovian).

**Behavior Cloning**  The most straightforward approach in IL is BC, which assumes access to pre-collected demos $D = \{(s_t, a_t)\}_{t=1}^N$ generated by expert policies and learns the optimal policy with direct supervision by minimizing the BC loss $\mathbb{E}_{(s,a) \sim D}[-\log \pi(a|s)]$ w.r.t. a mapping $\pi$. It requires the learned policy to generalize to states unseen in the demos since the distribution $P(\tau_\pi)$ will be different from the demo one $P(\tau_D)$ at test time, a challenge known as distribution shift [68]. Recently, several methods, particularly those based on Transformers, are proposed to relax the Markovian assumption. Instead of $\pi(a_t|s_t)$, the policy represents $\pi(a_t|s_{t-1}, s_{t-2}, ..., s_{t-T})$ or $\pi(a_t|s_{t-1}, a_{t-1}, ..., s_{t-T}, a_{t-T})$, i.e., considers the history up to a context size $T$. This change was empirically shown to be advantageous.

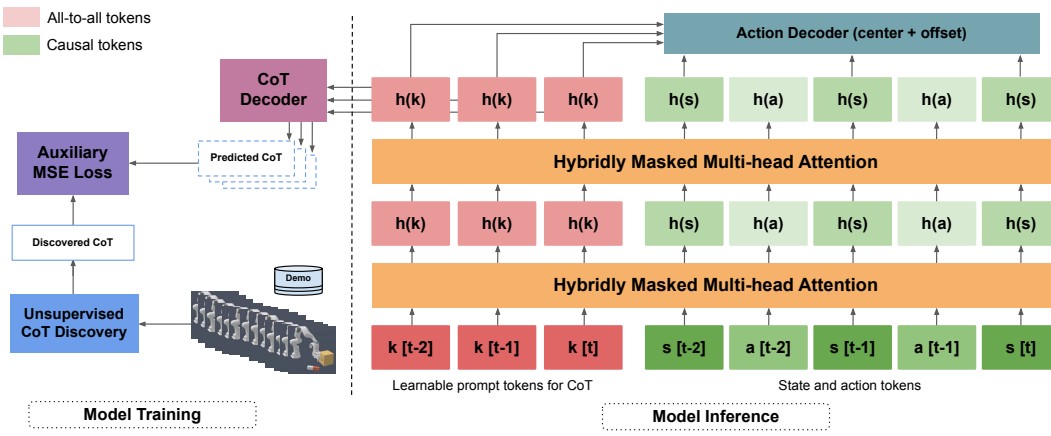

Figure 1: During training, CoTPC learns to jointly predict (1) the next & the last subgoals from each CoT token and (2) the low-level actions from each state token. During inference, without the CoT decoder, the low-level actions are predicted with the center & offset decoders from different tokens (detailed in Sec. 4.2.1) with the guidance of the dynamically updated CoT features. The CoT tokens are all-to-all (can see any tokens). The state and action tokens are causal (can only see previous and CoT tokens). Only 2 attention layers and 3 timesteps are shown for better display.

## 4 METHOD

To develop a scalable and powerful imitation learning algorithm for diverse yet sub-optimal demonstrations, especially for low-level control tasks, we propose (1) an observation space-agnostic strategy that efficiently discovers the subgoal sequences from the demonstrations (as chain-of-thought supervision) in an unsupervised manner and (2) a novel Transformer-based design that effectively adopts the hierarchical principles, the CoT planning (i.e., subgoal sequence predictions), in imitation learning.

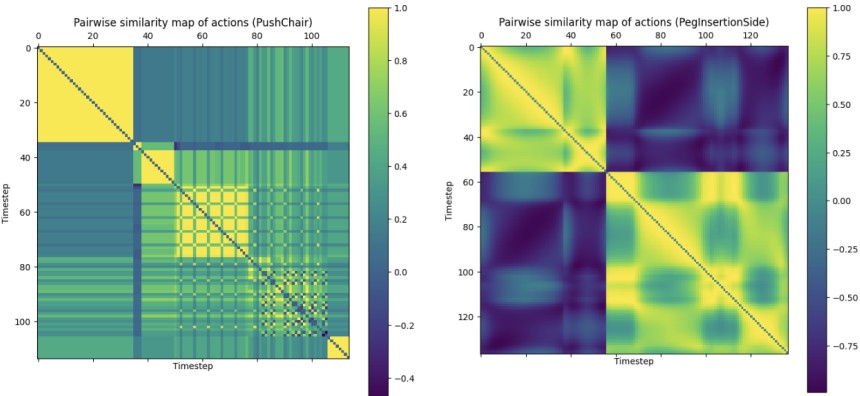

Figure 2: Pairwise similarities of actions at different timesteps in two trajectories for two example tasks, Push Chair (**left**) and Peg Insertion (**right**). Note that the action spaces for these tasks are distinct: the former uses delta joint velocity control and the latter uses delta joint pose control. Visually identifiable blocks along the diagonal, which have high correlation values among most of its members, tend to correspond very well with human intuition of subskills (i.e. similar actions at nearby timesteps belong to the same subskill). We find that this strategy allows for automatic, scalable, and meaningful subgoal discovery in an unsupervised manner.

## 4.1 Unsupervised Discovery of Chain-of-Thought from Demonstrations

We observe that many low-level control tasks (e.g., object manipulations) naturally consist of sequences of subgoals. In a succeeded trajectory, there exist key observations, each of which marks the completion of a subgoal. For instance, in Moving Maze illustrated in Fig. 3, the two bridges naturally divide the task into three subgoals. We denote such a multi-step subgoal decomposition of a task into a chain of planning steps as its chain-of-thought. The CoT provides coherent and succinct behavior guidance - typical benefits of hierarchical policy learning. Formally, for each trajectory $\tau \in D$, we define CoT as a sequence of its obervations $F_{cot}(\tau) = \{s_t | s_t \in \tau\} = \{s_k^{\mathbf{cot}}\}$. In this section, we present an intuitive strategy to discover CoT from demos as the CoT supervision for our model introduced shortly. Our unsupervised approach is observation space-agnostic and relies on neither human supervision nor reward design.

First of all, each demo trajectory can be considered as an execution of a chain of subskills to complete the corresponding sequence of subgoals. Accordingly, the ending observations of each subskill constitute the subgoal sequence (i.e., chain-of-thought). Also, note that the number of these subskills is not known *a prioi*, since demonstrations of the same task can still have variability in terms of the starting configurations and execution difficulty. Our base assumption is that actions within the same subskill are temporally close and functionally similar. We, therefore, propose to *group contiguous actions into segments*, using a similarity-based heuristic to find these subskills. We provide visualization of similarity maps across actions in the same trajectory in Fig. 2 to illustrate that there naturally exists such grouping patterns by functional similarities. While an action sequence is modeled as a time series, its subgoals are the key observations corresponding to the changepoints of the time series. We then utilize the Pruned Exact Linear Time (PELT) method [38] with cosine similarity as the cost metric to generate the changepoints in a per-trajectory manner. This unsupervised formulation has one key hyperparameter $\beta$ which controls the penalty for adding more changepoints thereby determining the number of subgoals (i.e., the length of the CoT). In our experiments, we select $\beta$ based on a small set of validation data and we find it relatively robust.

We find our approach discovers meaningful subskills (the action segments) for a diverse set of tasks with different action spaces, based on a qualitative evaluation of the key observations at the discovered changepoints (i.e., the chain-of-thought). For instance, an execution of Peg Insertion consists of reaching, grasping, aligning the peg with the hole, micro-adjusting and steady insertion of the peg. We illustrate some CoT discovery results in the Appendix. Besides being unsupervised, another major benefit of our approach is that it is observation space-agnostic with the discovery invariant of the specific sensor setup (e.g., camera angles).

## 4.2 Chain-of-Thought Guided Action Modeling

In this section, we introduce our Transformer-based design that learns to model the CoTs and predict the actions accordingly based on the supervision provided by the aforementioned CoT discovery.

### 4.2.1 Learnable Prompt Tokens for CoT with Hybrid Masking

We base our architecture on the recently proposed Behavior Transformer (BeT) [71], which empowers GPT [5] with a discrete center plus continuous offsets prediction strategy for modeling diverse and noisy action sequences. Please refer to Appendix A for a brief introduction to the BeT architecture. To predict CoTs, we propose to add a set of learnable prompt tokens [92] at the beginning of the state and action context history. We train these tokens to extract features from the sequence to predict CoTs together with a CoT decoder (similar to the object query tokens in Detection Transformer [6]). We design a *hybrid* masking regime, where during inference, the CoT tokens are all-to-all and can observe all action and state tokens in the context history, and the state or action tokens attend to those in the past (standard causal mask) including the CoT tokens. In this way, the action decoding is guided by the extracted CoT features. Formally, given a context size of $T$, let us denote the CoT tokens as $\{\mathbf{S}_{\cdots}^{\mathbf{cot}}\}$ (we will detail the subscripts in the next section). We model the demo trajectory segment of length $T$ up to timestep $t$, i.e., $\tau_T(t) = \{\mathbf{S}_{\cdots}^{\mathbf{cot}}, s_{t-(T-1)}, a_{t-(T-1)}, ..., s_{t-1}, a_{t-1}, s_t\}$ by applying the hybridly masked multi-head attention, denoted $\mathrm{MHA}_{hmask}[\cdot]$. Features from a total of $J$ attention layers are

$$h_j(\tau_T(t)) = \mathrm{MHA}_{hmask}[F_{enc}(\tau_T(t))], \quad j = 1$$
$$h_j(\tau_T(t)) = \mathrm{MHA}_{hmask}[h_{j-1}(\tau_T(t))], \quad j > 1$$

where $F_{enc}$ encodes each action token and state token by encoder $f_a(\cdot)$ and $f_s(\cdot)$, respectively (no encoder for the CoT tokens). Here we omit the position embeddings and the additional operations between the attention layers as in standard Transformers. Note that we also put the action sequences into the context in our variant of BeT and our model as we find this generally performs better.

### 4.2.2 Coupled Action and CoT Predictions with Shared Tokens

BeT adopts a pair of centers and offsets decoders for predicting actions, denoted as $g_a^{\mathbf{ctr}}(\cdot)$ and $g_a^{\mathbf{off}}(\cdot)$, with features from the last attention layer corresponding to the state tokens as inputs, which are denoted as $h_J(\tau_T(t))\mathtt{[-(2T-1)::2]}$ (with the Python slicing notation). At the center of our proposed approach is a CoT decoder $g_{cot}(\cdot)$, whose inputs are the last attention features from the CoT tokens $h_J(\tau_T(t))\mathtt{[:T]}$, to predict the CoT. Specifically, we couple action and CoT predictions by letting the offset decoder of actions $g_a^{\mathbf{off}}(\cdot)$ take features from the CoT tokens as inputs rather than from the state tokens as in the original BeT. With $g_a^{\mathbf{off}}(\cdot)$ and $g_{cot}(\cdot)$ sharing the same inputs, our architecture provides stronger CoT guidance for action modeling. We use $T$ CoT tokens, denoted $\{\mathbf{S}_i^{\mathbf{cot}}\}$ with $i \in \{0, 1, ..., T-1\}$, so that each token's features are used to predict the offset components of actions at one of the $T$ corresponding timesteps. See Fig. 1 for an overall architecture illustration and see Fig. 5 in the Appendix for detailed data flow regarding the action decoders (both offset and center) and the CoT tokens.

We formulate an autoregressive prediction strategy for CoT by training $g_{cot}(\cdot)$ to decode the next subgoal (i.e., the first $s_k^{\mathbf{cot}}$ with $k > t$ for the demo segment $\tau_T(t)$) and the very last subgoal (usually the end of $\tau$) from every CoT token. Formally, for each CoT token $\mathbf{S}_i^{\mathbf{cot}}$, we learn to predict $g_{cot}(h_J(\tau_T(t))\mathtt{[i]})$ as the next and the last subgoals. We have $T$ predictions for the same CoT outputs so that each CoT token learns to extract useful features for subgoal planning regardless of which timestep's action offset it is responsible for predicting. We find this strategy outperforms predicting only the immediate or the last subgoal, as it performs both immediate and long-term planning. The flexible numbers of subgoals across different demo trajectories necessitate the autoregressive decoding of CoTs and make our approach resonate with CoTs in LLMs, where similarly the outputs are generated jointly with the reasoning chain. Note that during inference the CoT decoder is not used and only the CoT features are. As CoT features are updated dynamically, our approach can deal with tasks involving dynamic controls (e.g., Moving Maze and Push Chair).

An alternative design of our approach is to only use one CoT token and use CoT predictions as $g_{cot}(h_J(\tau_T(t))\mathtt{[0]})$. In this case, the offset prediction of actions is based only on the state tokens.

However, we find this decoupled strategy produces policies much less generalizable. An even more coupled prediction approach is to force the center decoder $g_a^{\mathbf{ctr}}(\cdot)$ to also share the inputs with $g_a^{\mathbf{off}}(\cdot)$ and $g_{cot}(\cdot)$. However, this leads to optimization challenges and instabilities as CoT tokens hold direct responsibilities for all three predictions. See ablation studies for further discussions.

### 4.2.3 MODEL TRAINING

The overall training pipeline is illustrated in Fig. 1. The model is trained with behavior cloning loss as well as the auxiliary CoT prediction loss $\mathcal{L}_{cot}$ based on MSE (weighted by a coefficient $\lambda$), which yields the overall training objective:

$$\mathcal{L}_{total} = \mathop{\mathbb{E}}_{(s_t, a_t) \in D} \mathcal{L}_{bc}(a_t, \hat{a}_t) \; + \; \mathop{\mathbb{E}}_{\tau' \in D} \frac{1}{T} \sum_i \mathcal{L}_{cot}([s_{next}^{\mathbf{cot}}(\tau'), s_{last}^{\mathbf{cot}}(\tau')], g_{cot}(\mathbf{S}_i^{\mathbf{cot}}))$$

Where $\hat{a}_t$ is the predicted action via $g_a^{\mathbf{ctr}}(\cdot)$, $g_a^{\mathbf{off}}(\cdot)$. $\tau'$ are randomly sampled segments of the demo trajectories of length $T$. $s_{next}^{\mathbf{cot}}(\tau')$, $s_{last}^{\mathbf{cot}}(\tau')$ are the next and last subgoal, respectively. Note that there are $T$ CoT tokens and so there are $T$ CoT loss terms.

During training, we apply random attention masks to the action and state tokens so that the CoT tokens attend to a context of varied length (from the first state token in the context to a randomized $i$-th state token with $i \in \{0, 1, ..., T-1\}$). In doing so, we can (1) prevent CoT tokens from directly copying the corresponding state tokens which makes offset prediction of actions trivial and (2) perform a way of data augmentation (applying a form of dropout on the attention masks).

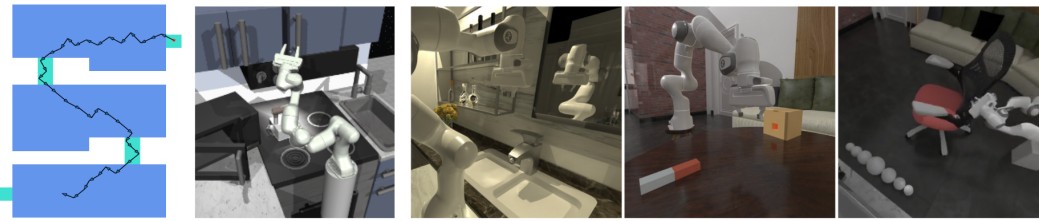

Figure 3: Illustration of the Moving Maze (**left**), Franka-Kitchen (**middle**) and some sampled tasks from ManiSkill2 (**right**), namely Turn Faucet, Peg Insertion and Push Chair. See detailed descriptions in Sec. 5.1, 5.2 and 5.3, respectively.

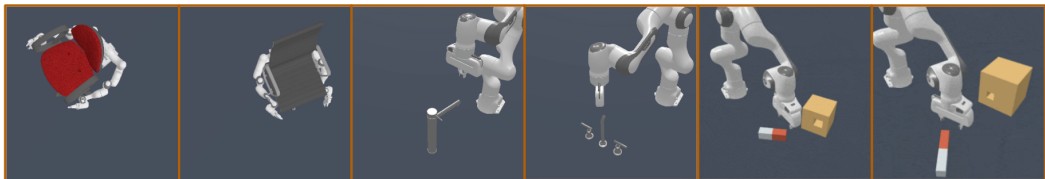

Figure 4: Sampled geometric variations for Push Chair, Turn Faucet and Peg Insertion. Note that the sizes of peg & box and the relative locations of the hole vary across different env. configs.

## 5 EXPERIMENTS

In this section, we present experimental results for several tasks as well as for the ablation studies. While existing benchmarks are mostly saturated for IL (DMControl [77], D4RL [22], etc.) or lack demo data (e.g., MineDojo [17]), we choose a diverse range of tasks that lie in between. We first examine our approach using a 2D continuous-space Moving Maze and a variant of Franka-Kitchen [26]. We then perform extensive comparisons with 5 object manipulation tasks (ranging from relatively easy ones to very challenging ones) from ManiSkill2 [25]. These tasks are of several categories (navigation, static/mobile manipulation and soft-body manipulation, etc.), various action spaces (delta joint pose, delta velocity, etc.), with different sources of the demos (human, heuristics, etc.) and with different reasons for being challenging (object variations, long-horizon, etc.).

Table 1: Test performance on Moving Maze and (a variant of) Franka Kitchen. SR (%) is the task success rate (for Franka Kitchen it means completion of all 4 sub-tasks). # s-tasks means the avg. number of completed sub-tasks per trajectory rollout. The best results are **bolded**.

|  | Vanilla BC | DT | BeT | CoTPC (ours) |
|---|---|---|---|---|
| Moving Maze (SR) | 9.0 | 23.0 | 33.0 | **44.0** |
| Franka Kitchen (#s-tasks/SR) | 1.7/6.7 | 1.6/6.7 | 1.8/14.4 | **2.1/25.6** |

## 5.1 Moving Maze

We present a 2D maze with a continuous action space of displacement $(\delta x, \delta y)$. As shown in Fig. 3 (left), in this s-shaped maze, the agent starts from a location randomly initialized inside the top right square region (in green) and the goal is to reach the bottom left one (also in green). Upon each environment reset, the two regions as well as the two rectangular bridges (in green) have their positions randomized. During the game, each of them except for the top square moves (independently) back and forth with a randomized constant speed. Once the agent lands on a moving block, the block will immediately become static. The agent cannot cross the borders of the maze (but it will not die from doing so). For simplicity, we adopt a 10-dim state observation consisting of the current location of the agent and the four green regions. This task requires *dynamic control/planning*.

**Demonstrations**   To enable policy learning from demonstrations, we curate demo trajectories (each with a different randomized environment configuration) by adopting a mixture of heuristics and an RRT-style planner with hindsight knowledge not available at test time (see details in the Appendix). This setup follows recent work [25] for leveraging machine-generated demonstrations.

**Training and Evaluation**   For this task, we compare CoTPC with vanilla BC, Behavior Transformer (BeT) [71], Decision Transformer (DT) [8]. DT was originally proposed for offline RL with demonstrations of dense rewards. We adapt it for the BC setup by ignoring the reward tokens. We add action tokens to BeT (like in DT) and build CoTPC on top of BeT. We implement CoTPC, DT and BeT with the shared Transformer configuration for a fair comparison. We train all methods on 400 demo trajectories of different env. configs and evaluate on 100 unseen ones (results in Tab. 1).

## 5.2 Franka Kitchen

Different variants and task setups of the Franka Kitchen environment have been studied previously [26, 70, 22]. We propose a setting where the agent is asked to complete 4 object manipulations (out of 7 different options) in an order specified by the goal. We use a strict criterion, i.e., the task succeeds when all 4 sub-tasks are completed. The sub-tasks need to be done in the requested order to be counted as completed. The environment will terminate when a sub-task other than the specified 4 is performed. The action space is based on the joint velocity (8-dim) of the robot. We use the original state observation appended with the modified goal embedding.

**Demonstrations**   We replay a subset of the human demonstrations originally proposed in [26]. Specifically, we use 50 demo trajectories of length ranging from 150 to 300 and relabel them with what sub-tasks are performed and in what order, for each of the trajectories. As a result, many ordered sub-task combinations admit at most one demo trajectory. See more details in Appendix.

**Training and Evaluation**   We use the same set of baselines as in Moving Maze. We evaluate using 90 unseen env. configs, which vary in initial scene configs (all ordered sub-task combinations have been observed in the demo, though). This task requires generalizable IL due to the limited amount of human demos and the diverse set of ordered sub-task sequences. Also see results in Tab. 1.

## 5.3 ManiSkill2

ManiSkill2 [25] is a recently proposed extension of ManiSkill [52], which features a variety of low-level object manipulation tasks in environments with realistic physical simulation (e.g., fully dynamic grasping motions). We choose 5 tasks (some illustrated in Fig. 3). Namely, Stack Cube for

Table 2: Test performance (success rate) on the unseen and the 0-shot setup for ManiSkill2 tasks with state observations. The best results are **bolded**. Note that Pour does not support state observations.

| | STACK CUBE | | PEG INSERTION | | TURN FAUCET | | PUSH CHAIR | |
|---|---|---|---|---|---|---|---|---|
| | UNSEEN | 0-SHOT | UNSEEN | 0-SHOT | UNSEEN | 0-SHOT | UNSEEN | 0-SHOT |
| VANILLA BC | 1.0 | 0.0 | 0.0 | 0.0 | 0.0 | 0.0 | | |
| DECISION TRANSFORMER | 19.0 | 17.5 | 40.0 | 27.0 | 25.6 | 17.0 | | |
| DECISION DIFFUSER | 26.0 | 12.6 | 17.0 | 5.0 | **56.0** | 20.0 | | |
| BEHAVIOR TRANSFORMER | 73.0 | 42.5 | 49.6 | 32.5 | 44.0 | 33.4 | | |
| CoTPC (OURS) | **86.0** | **59.3** | **50.0** | **39.3** | 51.2 | **41.0** | | |

Table 3: Test performance (success rate) on the unseen and the 0-shot setup for ManiSkill2 tasks for point cloud observations. The best results are **bolded**. We only show the best baseline here, i.e., BeT.

| | CUBE | | PEG | POUR | FAUCET | | CHAIR | |
|---|---|---|---|---|---|---|---|---|
| | UNSEEN | 0-SHOT | UNSEEN | UNSEEN | UNSEEN | 0-SHOT | UNSEEN | 0-SHOT |
| BEHAVIOR TRANSFORMER | 70.0 | 35.0 | 24.0 | 50.0 | 20.0 | 26.0 | 13.4 | |
| CoTPC (OURS) | **81.0** | **44.0** | **32.0** | **58.0** | **27.5** | **32.0** | **16.7** | |

picking up a cube, placing it on top of another, and the gripper leaving the stack; Turn Faucet for turning on different faucets; Peg Insertion for inserting a cuboid-shaped peg *sideways* into a hole in a box of different geometries and sizes; Push Chair for pushing different chair models into a specified goal location (via a mobile robot); and Pour for pouring liquid from a bottle into the target beaker with a specified liquid level. Push Chair adopts a delta joint velocity control (19-dim, dual arms with mobile base); Pour adopts delta end effector pose control (8-dim); the rest uses delta joint pose control (8-dim). We perform experiments with both state and point cloud observations.

**Task Complexity** The challenges of these tasks come from several aspects. Firstly, all tasks have all object poses fully randomized (displacement around 0.3m and $360°$ rotation) upon environment reset (this is in contrast to environments such as Franka Kitchen). Secondly, Turn Faucet, Peg Insertion and Push Chair all have large variations in the geometries and sizes of the target objects (see illustrations in Fig. 4). Moreover, the faucets are mostly pushed rather than grasped during manipulation (under-actuated control), the holes have 3mm clearance (requiring high-precision control) and it needs at least half of the peg to be pushed sideway into the holes (harder than similar tasks in other benchmarks [85]), the chair models are fully articulated with lots of joints, and the pouring task requires smooth manipulation without spilling the liquid. Moreover, ManiSkill2 adopts impedance controllers that admit smoother paths than the position-based (or specifically-tuned) ones while at the cost of harder low-level action modeling (e.g., actuators can be quite laggy).

**Demonstrations** The complexity of the tasks also lies in the sub-optimality of the demos (e.g., vanilla BC struggles on all 5 tasks). The demos here are generated by a mixture of multi-stage motion-planing and heuristics-based policies (with the help of privileged information in simulators). We use 500 demo trajectories for Stack Cube and Turn Faucet (distributed over 10 faucets), 1000 demos for Peg Insertion and Push Chair (distributed over 5 chairs) and 150 demos for Pour.

**Training and Evaluation** Besides vanilla BC, DT and BeT, we add Decision Diffuser (DD) [2] as a baseline, which explores the diffusion model [29] for policy learning by first generating a state-only trajectory and then predicting actions with an acquired inverse dynamics model. As tasks in ManiSkill2 feature diverse object-level variations, they provide good insights into both how effective an imitator can learn the underlying behavior and how generalizable it is. We evaluate using the 5 tasks in both unseen (seen objects but unseen scene configs) and 0-shot (unseen object geometries) setup. Specifically, we have all but Peg Insertion with the unseen setup and Turn Faucet, Push Chair &

Table 4: Results from the ablation studies (unseen SR for Push Chair and 0-shot SR for Peg Insertion).

|  | DECOUPLED | ONLY LAST | ONLY NEXT | RANDOM | VANILLA | O-SHARED | SWAPPED | CoTPC |
|---|---|---|---|---|---|---|---|---|
| PEG INSERTION | 47.0 | 52.0 | 49.0 | 41.0 | 45.0 | 39.0 | 46.0 | 59.3 |
| PUSH CHAIR | 36.0 | 36.0 | 37.0 | 31.0 | 35.0 | 29.0 | 32.0 | 41.0 |

Peg Insertion with the 0-shot setup. We use task success rate (SR) as the metric. Results are reported in Tab. 2 for state observation and Tab. 3 for point cloud observations, where we demonstrate the clear advantages of CoTPC. Note that we only select the best baseline (i.e., BeT) for experiments with point cloud observations, where we use a lightweight PointNet [61] for encoding the point clouds. For the main results of all methods on ManiSkill2, we report the best performance among 3 training runs over the last 20 checkpoints, an eval protocol adopted by [11].

## 5.4 ABLATION STUDIES

We present ablation studies on two tasks (Peg Insertion and Push Chair) from ManiSkill2 (we use state observations). We summarize the results in Tab. 4 and introduce the details below.

**Decoupled prediction of CoT and actions** In this variant denoted `decoupled`, we train another Transformer (denoted CoT Transformer) with the same state and action sequence as inputs to predict the CoT (the next and the last subgoal). While the original Transformer learns to only predict the actions with the discovered CoT fed into the CoT tokens. During inference, the CoT predicted by the CoT Transformer is fed instead. This decoupled CoT and action prediction strategy is inferior to the coupled one since the latter not only learns to leverage predicted CoT as guidance but also encourages better feature representation of the trajectory by sharing the features for these tasks.

**What subgoals to predict from the CoT tokens?** In the variant named `only last`, we ask the CoT decoder $g_{cot}(\cdot)$ to only predict the last subgoal. In the variant named `only next`, we ask it to only predict the next subgoal. We find that predicting both works the best as it provides the model with both immediate and long-term planning. In the variant named `random`, we ask it to predict a single randomly selected observation from the future as a random subgoal (not the ones from our CoT discovery). This leads to the worst results as the action guidance is not very helpful.

**Shared tokens for CoT and action predictions** A vanilla design of jointly predicting CoT and actions is to only use one CoT token with the center and offset decoders on top of the state tokens and the CoT decoder on top of the CoT token, denoted as `vanilla`. We further force both the center and the offset decoders to take CoT tokens as inputs, denoted `o-shared`. This variant overly shares the CoT tokens for all decoders and suffers from optimization instabilities. In the last variant, denoted `swapped`, we let the offset decoder take the action tokens as inputs and the center decoder take the CoT tokens. We find this alternative setup performs worse. We illustrate the data flow of these variants in Appendix A.

## 5.5 ADDITIONAL EXPERIMENTS

We perform two experiments to explore the possibilities for sim-to-real transfer of state observation-based CoTPC. We present some promising preliminary results in the Appendix.

## 6 CONCLUSION AND DISCUSSION

In this work, we propose CoTPC, a hierarchical imitation learning algorithm for learning generalizable policies from scalable but sub-optimal demos. We formulate the hierarchical principles (in the form of chain-of-thought) in offline policy learning with a novel Transformer-based design and provide an effective way to obtain chain-of-thought supervision from demonstrations in an unsupervised manner. We demonstrate that CoTPC can solve a wide range of challenging low-level control tasks, consistently outperforming many existing methods.

Many existing work dealing with "long-horizon" robotic tasks (SayCan [1], ALFRED [73], etc.) assume that low-level control is solved or that the task hierarchy is given. On the contrary, in this paper, we study better ways to learn to solve low-level control tasks with unsupervisedly discovered hierarchical information as supervisions. We believe that CoTPC can be extended in a multi-task learning setup, i.e., policy learning from diverse demos across different low-level control tasks.

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

## A  CHALLENGES OF IMITATION LEARNING FROM SUB-OPTIMAL DEMONSTRATIONS FOR LOW-LEVEL CONTROL TASKS

**Non-Markovity**   While each trajectory in the demos can be represented by a Markovian policy, the Markovian policy linearly combined from them by perfectly imitating the combined demos can suffer from a negative synergic effect if there are conflicts across demos. This is because the demos might be generated by different agents or different runs of the same algorithm. It becomes even worse when the demonstrations themselves are generated by non-Markovian agents (e.g., human or planning-based algorithms). Instead, a non-Markovian policy is more universal and can resolve conflicts by including history as an additional context to distinguish between different demos.

**Noisiness**   Sometimes the demo trajectories are intrinsically noisy with divergent actions produced given the same states. For instance, a search-based planner returns more than one possible action given the same action and state history to solve the task. At times, the demo actions are even distributed uniformly (e.g., with motion planning algorithms as demonstrators). This leads to increased uncertainty and variance of the cloned policies and so higher compounding errors. Note that multi-modality is a related but orthogonal issue [71], i.e., when a unimodal estimate of the (continuous) action distribution leads to a significantly worse return.

**Discontinuity**   For low-level control tasks, demo policies often consist of sharp value changes or topology changes (e.g., due to contact changes). Such discontinuity in the underlying state-to-action mapping leads to difficulties in learning a robust and accurate model, thus harming generalizability. A recent method [20] deals with this by an energy-based implicit model in place of an explicit one. While theoretically sound, it is shown [71] to be less practical for non-Markovian implicit models, and several later non-Markovian explicit models outperform it.

**Randomness**   The actual or apparent unpredictability usually exists in sub-optimal demonstrations either because the intermediate computations of the demonstrators are not revealed in the demos (e.g., the shortest paths generated by BFS do not reveal the intermediate search process), or the demonstrators are inherently non-deterministic (e.g., relying on rejection sampling). Such a trait makes IL less robust as the decision-making patterns from demos might be unclear, hard to learn and not generalizable [56]. For instance, in a continuous action space maze, a solution found by random search is more-or-less a winning lottery ticket, whose pattern might not be very generalizable.

# B  A Brief Introduction to Behavior Transformer

To make our presentation self-contained, we include a brief introduction to the main idea and the architecture of Behavior Transformer (BeT), from which our framework is adapted. BeT adopts the standard Transformer architecture sequence modeling while aiming to handle the multi-modalities problems for high-dimensional continuous action sequences. It utilizes two action decoders (on top of the last attention layer features of each state token) instead of just one. The first decoder predicts only the discrete cluster center of the output action (a classification task). The second decoder predicts the offset corresponding to the cluster centers (a regression task). The final action output is the sum of the two (similar to the strategy commonly used in the object detection community). Specifically, BeT partitions the action space into different clusters by applying k-means to the actions recorded in the demonstration set. As it aims to sample from a continuous distribution (at test time), the offset decoder predicts not only the offset for the most likely cluster center but for all others (i.e., it predicts a vector of dimension `action dimension × num of cluster` every time). While in our experiments we do not sample actions (choose the most likely cluster instead) during inference for all methods being compared, we keep this design of predicting offsets of all clusters as we find it eases model learning.

# C  Illustration of the Ablation Study on Network data flow

To better explain the difference among the variants in our third ablation study in Sec. 5.4 (i.e., regarding "shared tokens for CoT and action predictions"), we illustrate the network data flow of the three variants as well as the original design in Fig. 5.

# D  Additional Discussions of Related Work

**Procedure Cloning (PC)**   PC [86] was recently proposed to use intermediate computation outputs of demonstrators as additional supervision for improving the generalization of BC policies. However, it assumes full knowledge of the demonstrators, including the usually hidden computations that consist of potentially large amounts of intermediate results. For instance, in the graph search example used in the original paper, PC requires knowing the traversed paths of the BFS algorithm, such as the status of each node, either the included ones or *the rejected (and so omitted) ones* in the final returned result. Whereas, CoTPC does not require such knowledge, as the CoT are included as part of the results, not hidden intermediate computations, and the CoT supervision itself can be obtained via the unsupervised discovery method. Moreover, machine-generated demonstrations can be crowd-sourced and the demonstrators are usually viewed as black boxes, making this a limitation of PC.

**Policy Learning with Motion Planning**   Some existing work [72, 32] adopt a strategy similar to CoTPC in terms of predicting key states (the waypoints) as high-level policies. However, they use motion planners as low-level policies, while CoTPC directly learns to predict low-level controls. The major advantage of our approach is to handle environments requiring dynamic (or reactive) controls (like Moving Maze and Push Chair), where the key states must be updated at every step and so motion planner-based strategies will struggle. Also, PerAct [72] is relatively limited due to its discretized actions especially for tasks requiring high-precision manipulation (e.g., Peg Insertion). Moreover, some existing work [63, 19] use motion planners as the only demonstrators to acquire neural policies, which are to some extent constrained to tasks involving quasi-static control.

**Robotic Transformer-1 (RT-1)**   RT-1 [4] is a concurrent work that also directly models low-level control actions with a Transformer. It benefits from the sheer scale of real-world robot demonstration data pre-collected over 17 months and the tokenization of both visual inputs (RGB images) and low-level actions. While RT-1 shows great promise in developing decision foundation models for robotics, it adopts the conventional auto-regressive Transformer without explicitly leveraging the structural knowledge presented in low-level control tasks. Moreover, it is so computationally intensive that it usually only admits less than 5 control signals per second. Our work, CoTPC, is an early exploration in this direction and we believe it will inspire the future designs of generally applicable models for robotics tasks. Another difference is that since RT-1 discretizes the action space, it might suffer from degraded performance for tasks that require high precision (such as Peg Insertion).

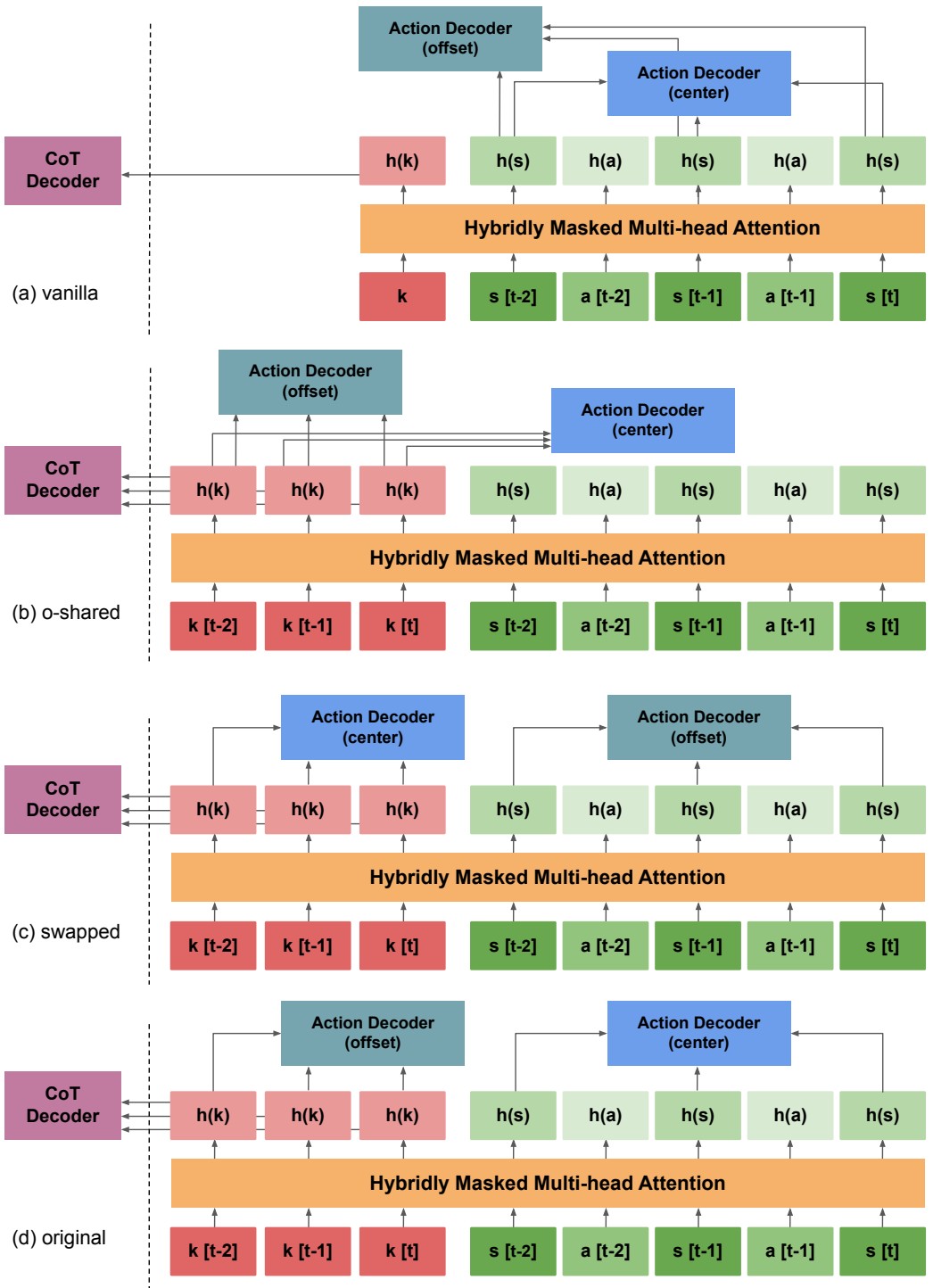

Figure 5: Illustration of the network data flow of our third ablation study for the variant named `vanilla`, `o-shared`, and `swapped` as well as the original design.

# E    Details of the Environments

**Moving Maze**    Moving Maze is a 2D maze with a continuous action space of displacement $(\delta x, \delta y)$, where both components $\in [1.5, 4]$. This an s-shaped maze whose height is 80 and width is 60 with the agent starting from a location randomly initialized inside the top right square region (in green) and the goal is to reach the bottom left one (also in green). Upon each environment reset, the two regions (the starting square and the target square) as well as the two rectangular bridges (in green) have their positions randomized. Specifically, The two square regions are randomized to the right of the top and to the left of the bottom blue islands, respectively. Their initial locations vary with a range of 20 vertically. The two bridges' initial locations vary also with a range of 20 horizontally. During the game, each of them except for the top square moves (independently) back and forth with a randomized constant speed $\in [1, 2]$. Once the agent lands on a moving block, the block will immediately become static. The agent cannot cross the borders of the maze (but it will not die from doing so). For simplicity, we adopt a 10-dim state observation consisting of the current location of the agent and the four green regions. This task requires dynamic controls/planning.

**Franka Kitchen**    We propose a setting (and thus a variant of the original Franka Kitchen task) where the agent is asked to complete 4 object manipulations (out of 7 different options) in an order specified by the goal. The 7 tasks are: turn on/off the bottom burner, turn on/off the top burner, turn on/off the light, open/close the slide cabinet, open/close the hinge cabinet, open/close the microwave oven, and push/move the kettle to the target location. Compared to the other variants, we use a strict criterion, i.e., the task succeeds when all 4 sub-tasks are completed, where each of them needs to be done in the requested order to be counted as completed. The environment will terminate when a sub-task other than the specified 4 is performed. The action space is based on the joint velocity (8-dim) of the robot. We use the original 30-dim state observation consisting of poses of all the relevant objects and the proprioception signals as well as an additional 14-dim goal embedding. This embedding assigns a 2-d vector to each of the 7 potential sub-tasks. Each vector is one of $[0, 0], [0, 1], [1, 0], [1, 1]$ (indicating the order to be completed for the corresponding sub-task) and $[-1, -1]$ (meaning the sub-task should not be completed). Note that we do not include the target pose of the objects in the state observation (i.e., we ask the agent to learn it from the demonstrations).

**ManiSkill2**    ManiSkill2 [25] is a recently proposed comprehensive benchmark for low-level object manipulation tasks. We choose 5 tasks as the testbed. (1) Stack Cube for picking up a cube, placing it on top of another, and the gripper leaving the stack. (2) Turn Faucet for turning on different faucets. (3) Peg Insertion for inserting a cuboid-shaped peg sideways into a hole in a box of different geometries and sizes. (4) Push Chair for pushing different highly articulated chair models into a specified goal location (via a mobile robot). (5) Pour for pouring liquid from a bottle into the target beaker with a specified liquid level. All tasks have all object poses fully randomized (displacement around 0.3m and 360° rotation) upon environment reset (this is in contrast to environments such as Franka Kitchen). Note that the holes in Peg Insertion have only 3mm of clearance, requiring highly precise manipulation, and it needs at least half of the peg to be pushed sideway into the holes (in contrast to the similar yet easier tasks [85]). The tasks we select involve both static and mobile manipulation and cover 3 action spaces (delta joint velocity control for Push Chair, delta end-effector pose control for Pour and delta joint pose control for the rest). For state observations, we use states of varied dimensions across these tasks (see details in the ManiSkill2 paper) For Turn Faucet, we slightly modify the default state observation by appending an extra 3-dim vector (the pose of the faucet link) so that it is easier for the agent to distinguish between different faucet models. The corresponding demonstrations are modified as well. For point cloud observations, we use the default pre-processing strategy provided by ManiSkill2 to obtain a fixed length of 1200 points (RGB & XYZ) per timestep.

# F    Details of the Demonstrations and the Evaluation Protocol

**Moving Maze**    We curate demonstrations by adopting a mixture of heuristics and an RRT-style planner with hindsight knowledge not available at test time. For each randomized environment configuration, we randomly choose one of the found paths from the starting square to the target one as the demonstration trajectory. We chunk the maze into 6 regions: the three islands (each bridge belongs to the island below it; the starting square and the target square belong to the top right and the bottom left regions, respectively), each of which is further divided into two regions (cut vertically in

the middle). An RRT-style sampler is used to find paths connecting adjacent regions sequentially (starting from the initial position to the second region). We restrict the number of steps in each of the paths across two adjacent regions to be $\leq 13$ so that the maximum total length of a demo trajectory is $\leq 13 \times 5 = 65$ steps. To enable this type of planning with dynamic environments, we actually first generate demonstrations with a static version of the maze and then animate the moving elements later coherently. This is not possible during inference time as it requires hindsight knowledge. We use 400 demo trajectories for training and evaluate all agents on both these 400 configs and a held-out set of 100 unseen environment configs. We report the task success rate as the major metric. During inference, we set the maximum number of steps as 60.

**Franka Kitchen**   We replay a subset of the human demonstrations originally proposed in [26] in the simulator. Specifically, we randomly select 50 demo trajectories of length ranging from 150 to 300 that succeed in achieving 4 different sub-tasks out of the 7 options. We relabel them with the privileged information to construct the goal embedding described previously. Note that this embedding vector is fixed across different time steps for each individual trajectory. There are 20 total different ordered sub-task combination presented in the 50 demonstrations, where the majority of combination only has $\leq 3$ trajectories. Combinatorial generalization regarding sub-tasks is too challenging in this case (there are $35 \times 4! = 840$ total combinations); so we focus on evaluating generalization w.r.t. initial robot/object poses. We use 90 unseen environment configurations (each requiring the completion of 4 sub-tasks) presented in the original human demonstrations for evaluation (we only include seen sub-task combinations). We report the task success rate (requiring the completion of all 4 sub-tasks in a trajectory) and the average number of successful sub-tasks per trajectory as the metrics. During inference, we set the maximum number of steps as 280.

**ManiSkill2**   For all tasks except for Push Chair, we use the original demonstrations provided by ManiSkill2, which are generated by a mixture of TAMP solvers and heuristics. Please see the original paper for details (the actual code used to generate these demonstrations is not released, though). For Stack Cube and Turn Faucet, we randomly sampled 500 demo trajectories for the training data. For Turn Faucet, we use trajectories from 10 different faucet models for the demos and perform the evaluation of 0-shot generalization on 4 unseen faucets (each with 100 different scene configurations). Not that the demonstrations of Turn Faucets have most of the faucets pushed rather than grasped, i.e., under-actuated control. For Peg Insertion, each different environment config comes with a different shape and size of the box (with the hole) and the peg. We randomly sampled 1000 trajectories from the original ManiSkill2 demonstrations as the training data and sampled 400 unseen environment configs for the 0-shot generalization evaluation. For Push Chair, we use a complicated heuristic-based approach from [55], adapted to ManiSkill2 (where it uses additional privileged information and achieves around 50% task success rate), as the demonstrator. We collected 1000 successful trajectories as the training data across 6 chair models and we evaluated the 0-shot generalization performance on 300 environment configs distributed over 3 unseen chair models. For Pour, we use 150 successfully replayed demonstration trajectories provided by ManiSkill2 to generate the point cloud observation sequences (this task does not support state observation as it involves soft-body). We use the success conditions from the original ManiSkill2 paper to report the task success rate. During inference, we set the maximum number of steps allowed as 200, 200, 200, 250 and 300 for Stack Cube, Push Chair, Peg Insertion, Turn Faucet and Pour, respectively.

## G   ILLUSTRATION OF THE DISCOVERED COT

In Fig. 6, we visualize the sub-stages (and thus the subgoals) extracted from two trajectories using the automatic CoT discovery process described in Sec. 4.1.

## H   IMPLEMENTATION DETAILS OF NETWORK ARCHITECTURE AND TRAINING

**Vanilla BC**   : We use a Markovian policy implemented as a three-layer MLP with a hidden size of 256 and ReLU non-linearity. We train it with a constant learning rate of $1e - 3$ with Adam optimizer with a batch size of 32 for 150K iterations (Moving Maze), 300K iterations (Franka Kitchen) and 500K iterations (ManiSkill2). We find training longer leads to over-fitting even with L2 regularization.

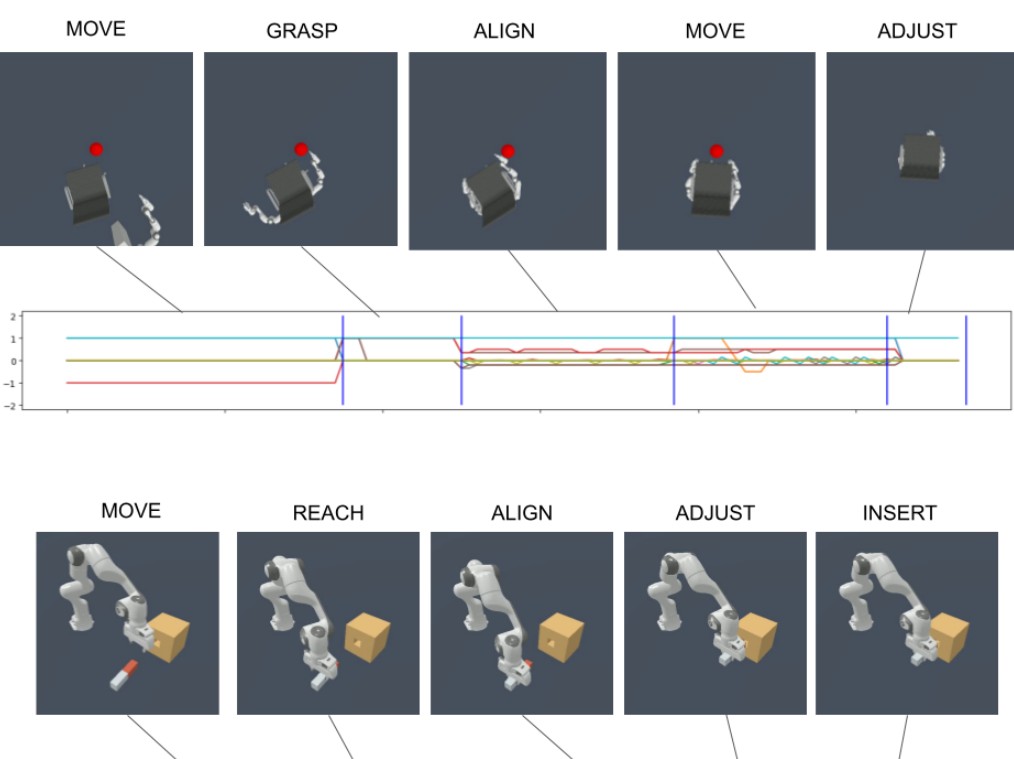

Figure 6: Illustration of actions corresponding to different stages and the associated observations for two tasks: Push Chair (**top**) and Peg Insertion (**bottom**). The stages are discovered by grouping the actions into subskills by our unsupervised CoT discovery method.

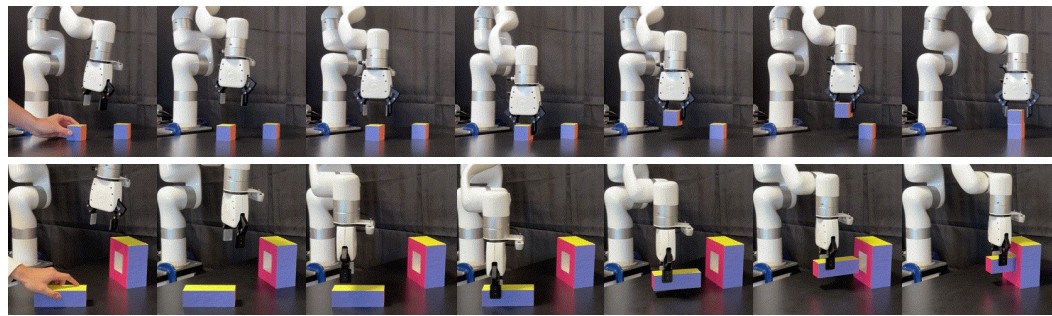

Figure 7: Two sampled succeeded trajectories for Stack Cube and Peg Insertion, respectively, in a real robot setup, from state-based CoTPC policies trained purely from demos in simulators. As an early examination, we increase the clearance for peg insertion from 3mm (sim) to 10mm (real) and only use peg and box-with-hole models of fixed geometry.

**Decision Diffuser**   We use the reference implementation provided by the authors of DD and make the following changes in the diffusion model: 100 diffusion steps, 20 context size, and 4 horizon length (in our experiments we found that longer performs worse). The diffusion and inverse-dynamics models have ∼1.6M parameters in total. Since DD works on fixed sequence lengths, we pad the start and end states during training and only the start states during inference.

**Decision Transformer**   We adopt the minGPT implementation and use the same set of hyperparameters for all tasks (a feature embedding size of 128 and 4 masked multi-head attention layers, each with 8 attention heads), totalling slightly greater than 1M learnable parameters. The action decoder is a 3-layer MLP of two hidden layers of size 256 with ReLU non-linearity (except that for Franka Kitchen we use a 2-layer MLP of hidden size $1024$). We train DT with a learning rate of $5e - 4$ with a short warm-up period and cosine decay schedule to $5e - 5$ for all tasks (except for Franka Kitchen, whose terminal lr is $5e - 6$ ) with the Adam optimizer with a batch size of 256. We train for 200K iterations for Moving Maze and Franka Kitchen and train for 500K iterations for all tasks in ManiSkill2. We use a weight decay of $0.0001$ for all tasks but Franka Kitchen (for which we use $0.1$). We use a context size of 10 for ManiSkill2 tasks, 20 for Moving Maze and 10 for Franka Kitchen. We use learnable positional embedding for the state and action tokens following the DT paper.

**Behavior Transformer**   We started with the configuration used for the Franka Kitchen task in the original paper. We changed the number of bins in K-Means to 1024 (we find that for our tasks, a smaller number of bins works worse) and changed the context size to 10 (in line with the other transformer-based models). The Transformer backbone has approximately the same number of parameters (∼1M) as CoTPC and DT. We train the model for around 50k iterations (we find that training longer leads to over-fitting easily for BeT, potentially because of its discretization strategy and the limited demos used for BC). For ManiSkill2 tasks, we use the same architecture as that for DT, except that we use the center plus offset decoders to decode the actions. We train 100K iterations for Turn Faucet, Push Chair and Pour. We train 200K iterations for Peg Insertion and Stack Cube.

**CoTPC**   Unless specified here, we keep other configurations (both model training and network architecture) the same as those in BeT. We use no positional embeddings for CoT tokens as they themselves are learnable prompts. The CoT decoder is a 2-layer MLP with ReLU non-linearity of hidden size 256. We use a coefficient $\lambda = 0.1$ for the auxiliary MSE loss for all tasks. During training, we apply random masking to the action and state tokens so that the CoT tokens attend to a history of varied length (from the first step to a randomized $t$-th step). Also see the main paper for more details.

## I    PRELIMINARY RESULTS OF SIM-TO-REAL TRANSFER

We examine the plausibilities of sim-to-real transfer of our state-based CoTPC in a zero-shot setup on two tasks, namely, Stack Cube and Peg Insertion. Our real-world experiment setup and two sampled

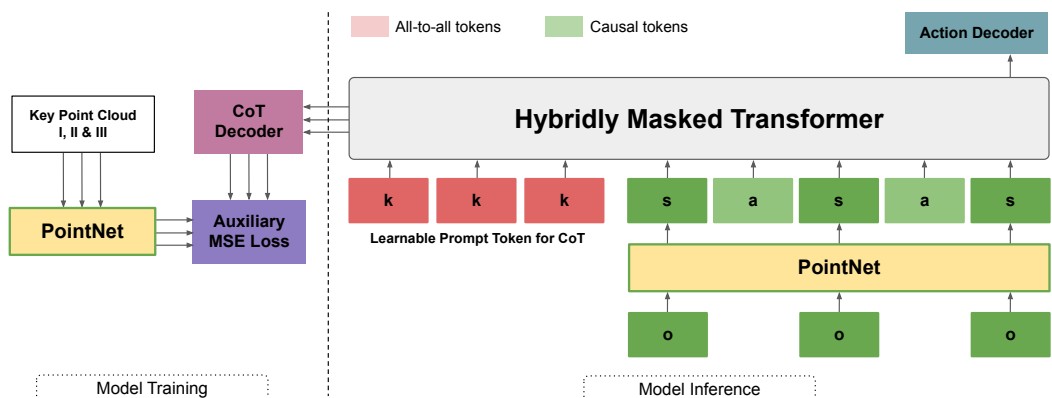

Figure 8: Illustration of the point cloud-based CoTPC. Compared to state-based CoTPC, we add a PointNet to process the point cloud observations as well as the point clouds CoT. We omit the data path for the input proprioception signals to the model.

succeeded trajectories are illustrated in Fig. 7. With an off-the-shelf pose estimation framework such as PVNet [57], we can achieve reasonable performance using the state-based CoTPC policy learned purely from simulated data. We also provide .gif animations as part of the supplementary materials. As a preliminary examination, we only perform qualitative evaluations.

## J   DETAILS OF POINT CLOUD-BASED CoTPC

To process point cloud observations, we adopt a lightweight PointNet [61] (∼27k parameters) that is trained from scratch along with the transformer in an end-to-end manner. We concatenate additional proprioception signals with the point cloud features to the input state tokens. In CoTPC where the CoT decoder is trained to predict the point cloud CoT, we ask the decoder to predict PointNet features of the CoT instead. We find that the auxiliary point cloud CoT loss causes the PointNet encoded representations to collapse. Inspired by [10], we use a stop-gradient operation in the point cloud CoT encoding path to prevent this. We illustrate the network architecture in Fig. 8. The training strategies (we use the same set of hyperparameters) and evaluation protocols are similar to those of the state-based experiments.

## K   ADDITONAL VIDEOS

We provide videos of inference results of our state-based CoTPC on ManiSkill2 tasks in the supplementary materials.

