# OpenReview forum: "Chain-of-Thought Predictive Control"
_ICLR.cc/2024/Conference — Submitted to ICLR 2024_

### Official Review · Reviewer_QpAT · 2023-10-14

**Soundness:** 2 fair
**Presentation:** 2 fair
**Contribution:** 2 fair
**Rating:** 5
**Confidence:** 4

**Summary:**

This paper proposes a policy learning from the demonstration method. The authors propose a novel hierarchical imitation learning that utilizes scalable demonstrations. The demonstration is decomposed into a sequence of key observations, and then CoT is leveraged to generalize policy learning.

**Strengths:**

1. I think the author's environment looks cool and real. This is very important in the policy learning domain and in the robotic domain. I see the supplementary video and I believe that this environment provides a good test environment for policy learning methods. The authors are encouraged by the reviewer to continue their study in this domain, even though this paper may be rejected.

2. The general direction of learning policy with demonstrations is good, and the hierarchical RL formulation for this task is also sound and important, although those two ideas are not very novel.

3. The numerical results are good and the improvement looks significant.

4. I like the supplementary video as well, although it can be improved (see below).

**Weaknesses:**

# 1. About the novelty

(1) I think CoT is not very novel in this context and I don't think the proposed approach can be regarded as CoT. The proposed approach refers to a demonstration split methodology (or a subgoal discovery mechanism in hierarchical RL). Since no language is involved, I don't think this method can be related to CoT.

(2) This paper is not positioned well in the context of hierarchical RL + demonstration, so the novelty is not well-stated. The authors should mention, discuss, or compare the following works [1-5]. Disclaim: I am not an author of any of these works. The idea of learning subgoals by similarity or diversity is not novel.

(3) Using transformers in policy learning is not novel as well, and I don't think the major goal of this paper is related to architecture design. If the goal is to claim the novelty of CoT control, the authors should try other architectures as well. Since nowadays transformers are very common architectures, I don't think this can be claimed as a major novelty.

(4) It's unclear why the authors use a single model to learn CoT and learn the action. The authors should try to compare to architectures like [6].

# 2. About the experiments

(1) The authors should discuss their results and summarize the conclusions in the main text. I'm confused about the main result in Tab. 1, and this is because the results are not discussed (the authors only say results in Tab.1 without further explanations). It seems that BC, DT, and BeT are not hierarchical RL methods. So it is very unfair. The authors should not compare to methods that do not use subgoals. Instead, the authors should mainly compare to hierarchical RL methods that leverage demonstrations. The authors should search for [1-5] as long as their follow-up works to get the most related hierarchical RL methods to compare to.

(2) The variance should be shown for each method, and the learning curves are required.

(3) The supplementary videos should be combined with a presentation video, which can reveal the comparisons between the proposed approach and previous works.

# 3. About the writing

(1) The function of the model is not discussed and section 4.2 looks confusing as a result. I think the function and signature of the model should be discussed prior to the model details.

(2) The details of the CoT algorithm are unclear. Particularly, these two sentences look very confusing to me.

"therefore, propose to group contiguous actions into segments, using a similarity-based heuristic to find these subskills." How does the group algorithm work? How to discretize continuous actions?

"We then utilize the Pruned Exact Linear Time (PELT) method [38] with cosine similarity as the cost metric to generate the changepoints in a per-trajectory manner." How does this work? I'm not familiar with the PELT method and this should be discussed in detail.

(3) The authors assume the readers know the decision transformer and detection transformer in advance, which is not very good.

[1] Jiang, Yiding, et al. "Learning Options via Compression." Advances in Neural Information Processing Systems 35 (2022): 21184-21199.

[2] Eysenbach, Benjamin, et al. "Diversity is all you need: Learning skills without a reward function." arXiv preprint arXiv:1802.06070 (2018).

[3] Konidaris, George, et al. "Robot learning from demonstration by constructing skill trees." The International Journal of Robotics Research 31.3 (2012): 360-375.

[4] Pickett, Marc, and Andrew G. Barto. "Policyblocks: An algorithm for creating useful macro-actions in reinforcement learning." ICML. Vol. 19. 2002.

[5] Kipf, Thomas, et al. "Compile: Compositional imitation learning and execution." International Conference on Machine Learning. PMLR, 2019.

[6] Target-driven Visual Navigation in Indoor Scenes using Deep Reinforcement Learning

**Questions:**

1. How does the number of CoT affect the policy learning results?

2. If the transformer's weights are tuned, why the CoT can accurately serve as a target signal?

3. How efficient is the transformer architecture?

---

> ### Author Response · Authors · 2023-11-22
> **Author Response to Reviewer QpAT**
>
> Thanks for the valuable feedback with so many details!
>
> **Since no language is involved, I don't think this method can be related to CoT.**
>
> We name our work with chain-of-though similar to [7], which suggests that CoT generally refers to the key intermediate steps in sequential decision-making. Nevertheless, we are willing to change the name to others (e.g., subskill-augmented or subskill-predictive control) if all of the reviewers think it is necessary (as the reviewer iFzy also raised this point).
>
> **Not positioned well in the context of hierarchical RL + demonstration regarding subgoal decomposition. See [1-5]**
>
> While the high-level idea of utilizing the hierarchical principles for policy learning is shared with the approaches in the hierarchical RL + demonstration literature (e.g., [1-4]), all of these work relies on online interaction with the environments either by using online RL to learn to utilize the learned (sub)skills in a transfer learning setup (e.g, [1]) or by using online exploration to acquire the (sub)skills in the first place (e.g., [2]). On the contrary, ours is a completely offline imitation learning method.
>
> Purely offline subskill discovery that can be used for downstream imitation learning is very challenging, and recent work mostly relies on RL to provide extra supervision [6]. (Also see the author's response to reviewer HDpH). Work that is similar to ours in this setup includes CompILE [5] and PC [7]. CompILE adopts a VAE to perform soft-segmentation of the demo trajectories. However, its evaluation was carried out only on a limited set of relatively simple control tasks. We try our best to find [8] as a relevant follow-up work, yet its evaluation only involves grid worlds. On the other hand, [7] performs hierarchical policy learning with extra procedure-level supervision that limits its applicability and is hard to be directly compared with (also see a discussion in our Appendix). Notice that due to the relatively small previous work in this direction (subskill discovery + hierarchical offline policy learning), both [5] and [7] only use non-hierarchical imitation learning methods as baselines. Similarly in our comparison, we include several non-hierarchical strong baselines.
>
> We will include a discussion regarding this in the appendix.
>
> **Since nowadays transformers are very common architectures, I don't think the architecture design can be claimed as a major novelty.**
>
> Our contributions are two-fold: a novel formulation of observation-agnostic subskill from offline demonstrations based on discovery based on breakpoint detection and a customized Transformer architecture for the subskill-augmented hierarchical imitation learning method.
>
> Our specific architecture design makes our framework suitable for leveraging the hierarchical information discovered in an offline manner. Granted, Transformers are common standard architectures nowadays. We do not aim for a major overhaul of Transformers (neither does existing work such as BeT). Nevertheless, our design to enhance Transformers for hierarchical imitation learning is shown to be effective and important with several ablations studies. Specifically, we add a diagram to the Appendix to better illustrate the differences in the network design in our third ablation study. We add an ablation study in the author's response to reviewer 2mSo in this direction.
>
> **Why do the authors use a single model to learn CoT and learn the action? Should try to compare to architectures like [9].**
>
> Here we add another ablation study to verify our architecture design. We report unseen SR for Push Chair and 0-shot SR for Peg Insertion. In this study, we show that using two separate models for predicting the CoT and the actions respectively is outperformed by the joint modeling approach. Note that the reviewer mentioned [9], yet it is not directly comparable since the setup of [9] takes target images as inputs rather than trying to predict them as outputs. Intuitively, the auxiliary loss we propose together with the joint learning approach improves sequential feature learning by encouraging the model to extract information predictive of not only the actions but also the hierarchical structure of the task.
>
> |    | CoTPC (sep) | CoTPC |
> | ------- | ------- | ------- |
> | Peg Insertion | 49.0 | 59.3 |
> | Push Chair | 35.7 | 41.0 |
>
> &nbsp;
>
> [1] Learning Options via Compression
>
> [2] Diversity is all you need: Learning skills without a reward function
>
> [3] Robot learning from demonstration by constructing skill trees
>
> [4] Policyblocks: An algorithm for creating useful macro-actions in reinforcement learning
>
> [5] Compile: Compositional imitation learning and execution
>
> [6] Hierarchical Imitation Learning with Vector Quantized Models
>
> [7] Chain of Thought Imitation with Procedure Cloning
>
> [8] Unsupervised Learning of Temporal Abstractions with Slot-based Transformers
>
> [9] Target-driven Visual Navigation in Indoor Scenes using Deep Reinforcement Learning

---

> > ### Comment · Reviewer_QpAT · 2023-11-22
> > **Thanks for your response.**
> >
> > Renaming this paper is good. And I think experimenting with online setups is necessary. Besides, ComPILE is not restricted to "relatively simple control tasks." It's a pretty general method, and I have seen or reviewed at least three drafts and papers using ComPILE. See https://scholar.google.com/scholar?cites=12302759254570528216&as_sdt=5,33&sciodt=0,33&hl=en.

---

> > > ### Author Response · Authors · 2023-11-22
> > > **Thanks for the additional feedback!**
> > >
> > > We agree that the online setup is very important. However, we believe it is out of the scope of this paper as offline policy learning and online policy learning have relatively different goals for solving hard control tasks based on the constraints of online interactions and the availability of offline demos. Although recently there are more approaches combining both, many works (DT, BeT, Diffusion Policy, PC, etc.) focus only on the offline setup. We will leave it as future work. We agree ComPILE is a rather general method and we have tried our best to search for the follow-up work on comPILE regarding the offline setup.
> > >
> > > Thanks again for the feedback.

---

> ### Author Response · Authors · 2023-11-22
> **Author Response to Reviewer QpAT (cont.)**
>
> **Need better writing for section 4.2 to introduce the details of the model**
>
> We updated Section 4.2 (mostly 4.2.2) to make it clearer (also see the author's response to the reviewer 2mSo).
>
> **The paper utilizes PELT for breakpoint detection. Might need to explain it in more detail as background knowledge. Should include an introduction to BeT as well.**
>
> We added a brief introduction paragraph to BeT to the appendix. We will add another short intro to the main paper for PELT. In short, PELT is an optimized version of a more classic breakpoint detection method that is based on the minimization of certain costs. A time series can be partitioned into consecutive groups where each group is associated with a cost term similar to the within-cluster distance in k-means. The total cost is the sum of all individual cost terms plus some penalty to control the number of groups. The classic breakpoint detection method performs dynamic programming to find the exact optimal partitioning that minimizes the total cost. PELT is an optimized version of the method.
>
> **Sensitivity of the length of CoT.**
>
> We perform experiments on the Peg Insertion task (state observation policies) and find that shorter CoT (approximately -1 length on average) leads to slightly decreased performance (59.3 vs. 54.0 in terms of 0-shot SR); we find longer CoT (approximately +1 length on average) have very similar performance.
>
> **The variance should be shown for each method, and the learning curves are required.**
>
> For all results on ManiSkill2, we report the best result among the three training runs in the original paper. We report the additional mean and std below. The format is best (mean ± std).
>
> |    | Stack Cube (unseen) | Peg Insertion (0-shot) | Turn Faucet (0-shot) | Push Chair (0-shot) |
> | ------- | ------- | ------- | ------- | ------- |
> | BeT | 73.0 (68.7±4.0)  | 42.5 (39.5±2.7) | 32.5 (31.7±0.8) | 33.4 (32.3±1.0) |
> | CoTPC | 86.0 (84.0±2.6) | 59.3 (51.5±7.4) | 39.3 (35.4±3.6) | 41.0 (36.4±4.0) |
>
> Note that the learning curves might not be very meaningful for purely offline imitation learning tasks. Both BeT and DT do not include them for the offline learning setup.
>
> **How efficient is the transformer architecture?**
>
> We use around 1 million parameters for all of the CoTPC, DT, and BeT models in our experiments. We wonder what additional information is requested here.
>
> **If the transformer's weights are tuned, why the CoT can accurately serve as a target signal?**
>
> We are not quite sure what is referred to here. The CoT tokens are learned to extract information predictive of the hierarchical structure of the task in the form of subgoals. The Transformer is learned to utilize the extracted information for better action decoding.

---

### Official Review · Reviewer_HDpH · 2023-10-28

**Soundness:** 3 good
**Presentation:** 3 good
**Contribution:** 2 fair
**Rating:** 5
**Confidence:** 3

**Summary:**

The authors present Chain-of-Thought (CoT) Predictive control, a transformer based approach to policy learning, trained via sequence modeling. The transformer is augmented with learnable CoT prompt tokens that guide low-level action learning. In addition, the transformer is trained to predict the next and last high-level prompt, further encouraging abstractions that capture higher level semantic information. The high-level prompts are discovered in an unsupervised manner, as changepoints in time, discovered with Pruned Exact Linear Time methods, using cosine similarity as a cost metric. The model is trained on suboptimal demos and surpasses other transformer based methods on held-out tasks.

**Strengths:**

The paper has several strengths:


1) Reasonably well written
2) Simplistic and effective approach that outperforms similar methods
3) Nice to see their approach applied to complex dynamic settings, and generalizing favorably

**Weaknesses:**

My main concern with the method is their motivation for how they perform sub-task decomposition. Cosine similarity metric seems heuristical, not well motivated, and anecdotal. Whilst the results are promising, it is unclear whether more principled decompositions  would lead to better results: e.g. obtained via bottleneck options [1] or gaussian processes [2]. The paper would benefit from a greater discussion/comparison on this front. It is unclear to me whether their decomposition approach would favor different tasks, with distinct action-space statistics.

In addition, there are a couple of presentation limitations:

1) Citations are not in the correct ICLR format (surname and year)
2) Results are lacking confidence intervals (how many runs/model seeds)

[1] - Salter, Sasha, et al. "Mo2: Model-based offline options." Conference on Lifelong Learning Agents. PMLR, 2022.

[2] - Saatçi, Yunus, Ryan D. Turner, and Carl E. Rasmussen. "Gaussian process change point models." Proceedings of the 27th International Conference on Machine Learning (ICML-10). 2010.

**Questions:**

1) How well does this approach to sub-task decomposition scale to larger action-spaces?
2) How sensitive is their approach to the beta parameter that controls the number of detected changepoints?
3) Fig 2 - Can the author's comment on what the action groupings correspond to intuitively for these examples?

---

> ### Author Response · Authors · 2023-11-22
> **Author Response to Reviewer HDpH**
>
> Thanks for the valuable review!
>
> **Major concern: Cosine similarity metric for sub-task decomposition seems heuristical. How does it scale up to a higher action dimension?**
>
> Our subskill discovery approach by decomposing the action sequences is motivated by the intuition that functionally similar actions that are temporally close should be grouped into the same subskill. The choice of the metric to measure such functional similarity is an open problem. We tried cosine similarity, L2 distance, and a more complicated Hausdorff distance. Among these, we find that the cosine similarity exhibits both simplicity and generalizability. In our experiments, we find it works across different action spaces (e.g., delta position in Moving Maze, delta joint pose in Peg Insertion, and delta joint velocity in Push Chair) with different action statistics based on how they are generated (sampling-based methods such as RRT, heuristics, etc.) and scale well to higher action dimensions (e.g., Moving Maze uses 2-d actions, Push Chair uses 19-d actions with a dual-arm mobile robot). In practice, action spaces of much higher dimensions are relatively uncommon in robotic tasks.
>
> **Discussion about the proposed decomposition approach compared to others in the literature (e.g, [1] and [2])**
>
> Skill or sub-skill discovery purely from offline demonstration sets is very challenging since there is barely any useful supervision. In the related literature, the option-based approach is a popular strategy. For instance, MO2 [1], an offline option learning framework using the bottleneck state principle, is shown to work well on continuous control problems for learning the options. However, this line of work has the shortcoming of relying on good state space representation. It is unclear if MO2 can work well for high-dimensional visual observations (e.g., the Pour task only supports visual observation due to soft-body manipulation). Our action-based approach is observation-agnostic and thus avoids this issue. Moreover, the option approach requires learning good initial and termination conditions for the options, which is a hard problem itself [2]. Usually, it requires further online learning for skill chaining to compensate for the suboptimal options (MO2 still requires online learning after the options learned from offline datasets are fixed).
>
> The alternative, as we propose, is to utilize methods from the breakpoint detection community for time series modeling to perform action segmentation. Breakpoint detection methods can roughly be divided into predictive model-based approaches (e.g., [3]) and optimization-based ones (e.g., [4]). The former requires an underlying predictive model (UPM) where [3] chooses Gaussian Processes. However, it is unclear if it can model high-dimensional and complex control signals as the action sequences can be hard to model in the first place ([3] suggests it might require a large set of sensitive hyper-parameters). The latter does not model the time series directly but finds breakpoints by optimizing a cost function instead. In this case, a good choice of a metric can be critical.
>
> We will add a discussion to the appendix.
>
> **Citations should be in ICLR format**
>
> We will fix it.
>
> **Results are lacking confidence intervals (how many runs/model seeds)**
>
> For all results on ManiSkill2, we report the best result among the three training runs in the original paper. We report the additional mean and std below. The format is best (mean ± std).
>
> |    | Stack Cube (unseen) | Peg Insertion (0-shot) | Turn Faucet (0-shot) | Push Chair (0-shot) |
> | ------- | ------- | ------- | ------- | ------- |
> | BeT | 73.0 (68.7±4.0)  | 42.5 (39.5±2.7) | 32.5 (31.7±0.8) | 33.4 (32.3±1.0) |
> | CoTPC | 86.0 (84.0±2.6) | 59.3 (51.5±7.4) | 39.3 (35.4±3.6) | 41.0 (36.4±4.0) |
>
> **How sensitive is their approach to the number of detected changepoints (length of CoT)?**
>
> We perform experiments on the Peg Insertion task (state observation policies) and find that shorter CoT (approximately -1 length on average) leads to slightly decreased performance (59.3 vs. 54.0 in terms of 0-shot SR); we find longer CoT (approximately +1 length on average) have very similar performance.
>
> **Fig 2 - Can the author's comment on what the action groupings correspond to intuitively for these examples?**
>
> Figure 6 in the appendix (updated pdf version) illustrates semantically what each subskill corresponds to.
>
> &nbsp;
>
> [1] Mo2: Model-based offline options
>
> [2] Multi-skill Mobile Manipulation for Object Rearrangement
>
> [3] Gaussian process change point models
>
> [4] Optimal detection of changepoints with a linear computational cost

---

### Official Review · Reviewer_iFzy · 2023-10-30

**Soundness:** 2 fair
**Presentation:** 3 good
**Contribution:** 3 good
**Rating:** 6
**Confidence:** 4

**Summary:**

This paper presents CoTPC, a behavior cloning method that predicts simultaneously multiple future sub-goals (Chain-of-thoughts), as well as low-level actions.  It also presents a method for discovering brakpoints and a chain of planning steps. It's evaluated on state-based tasks across various settings, from 2D moving maze, to franka kitchen, then to several tasks in maniskill2. The experiments show CoTPC outperforms other baselines as well as other ablation choices. It also show some preliminary results on 2 real world tasks.

**Strengths:**

I like the general direction this paper is pursuing. Addressing suboptimality in demonstrations by finding shared hierarchical patterns and key states makes a lot of sense. Predicting a sequence of subgoals simultaneously, as opposed to auto-regressively one-by-one, is also reasonable in terms of better guiding low-level actions prediction.
The paper has a set of extensive experiments, as well as some preliminary study using realistic visual inputs and real-world experiments.
In addition, i was a reviewer reviewing this paper during its previous round of submission, back then one of my major concern is lacking of a automated machanisim for extracting key states from the demo. This has been addressed to some extent in this version.

**Weaknesses:**

* I am still not fully convinced by using the term 'Chain-of-Thought'...
*  Real world evaluation is a bit too simple

**Questions:**

I have no further questions since I reviewed this before.

---

> ### Author Response · Authors · 2023-11-22
> **Author Response to Reviewer iFzy**
>
> Thanks for the valuable review (again)!
>
> Regarding the term CoT, we are willing to change the name to others (e.g., subskill-augmented or subskill-predictive control) if all of the reviewers think it is necessary (as the reviewer QpAT raised this point as well).

---

> > ### Comment · Reviewer_iFzy · 2023-11-22
> > **Reviewer response**
> >
> > Thank you for your response. Yes renaming would be a good move. I'll keep my current positive score.

---

### Official Review · Reviewer_2mSo · 2023-11-02

**Soundness:** 3 good
**Presentation:** 2 fair
**Contribution:** 3 good
**Rating:** 6
**Confidence:** 4

**Summary:**

This work proposes CoTPC, a Transformer-based architecture that performs hierarchical planning.
An important part of this method is the unsupervised discovery of subgoals, which assume that temporally close and similar actions belong to the same subskills.
The overall architecture uses learned subgoal embeddings and uses the goals discovered by the unsupervised algorithm to train these learned embeddings, as an auxiliary loss.
The remainder of the architecture fits into the family of the Behaviour transformer, with some design differences that impact the performance, and utilize the CoT learned embeddings.
Experiments on the Moving Maze, Franka Kitchen and ManiSkill2 environments show the effectiveness of CoTPC over baselines.
An ablation study is also presented to showcase the importance of different design choices in the architecture.

**Strengths:**

- The dataset chosen for experiments are relevant, and the results are quite convincing, with the CoT model clearly performing better than the baselines. I thought the exposition of the experiments section clear and easy to follow. The choice made on experiments were clear and, to my knowledge, the choice of baselines are fair.

- Although the architecture is limited in novelty, the method as a whole is novel and the specific design decisions are novel. I particularly liked the use of learned embeddings that are trained using an auxiliary loss with targets generated using an unsupervised subgoal discovery process.

- The proposed work is clearly motivated and properly positioned in the literature.

**Weaknesses:**

- The writing of section 4 needs improvements. Specifically, I found Section 4.2.2 quite hard to parse and easy to get lost. I would encourage the authors to re-write this section and redo Figure 1 so that things are clearer.
    - Specifically, I'm confused by what the inputs of $g_\text{CoT}$.
    - What is the CoT predictor? It appears once in the last paragraph of section 4.2.2.
    - Why aren't actions used as inputs to $g^x(.)$ functions?
    - What exactly are the contents of $\{ \mathbf{S}^\text{CoT}_{...}\}$? Do they change with time? What is the (...) subscript?
    - There seems to be $T$ CoT features, but that seems confusing as these are suppose to represent subgoals and are trained using the auxiliary $L_\text{CoT}$. I thought PELT was minimizing the number of goals. How do you ensure alignment between the learned tokens and the output of PELT?

- I found the ablation study to have limited value. I think it should aim to provide the reader with more intuition on what exactly is learned by CoT embeddings. This could possibly be shown on the maze environment, and would show clearly the discovery of the subgoals. I see Figure 5 in the appendix and it is a step in the right direction, but in my opinion, it would be be easier and more informative to show in the maze environment.

- As a minor point, I would ask the authors to express the limitations of the approach. For instance, is it always possible to assume that actions that are similar or close temporally belong to the same subskill?

**Questions:**

- It seems like BeT should have been explained in section 3 as the method seems to be heavily based on it. I leave that up to the authors to decide, but it could potentially ease the exposition. An alternative could be to present it in the appendix, similar to the Rt-1 paragraph.

- In the last paragraph of section 5.4 named "Shared tokens for CoT and action predictions", I think the different variants should be shown as a three part diagram two help the reader understand the differences in inputs. As of right now, I am not entirely sure what the differences are.

- I would be curious to see what the effect of setting the component of the auxiliary loss term to 0, but keeping learnable prompt tokens. I think this is basically equivalent to BC but with added learnable tokens , maintain the same capacity as the CoTPC architecture.

---

> ### Author Response · Authors · 2023-11-21
> **Author Response to Reviewer 2mSo**
>
> Thanks for the valuable review!
>
> **Section 4.2.2 writing needs improvement**
>
> 1. We’ve updated Section 4.2.2 to make it clearer.
> 2. We reword all “predictors” to “decoders” to eliminate ambiguities.
> 3. We describe the inputs of the CoT decoders in more detail.
> 4. Regarding why not use action tokens for predictions: The action tokens are not directly used as inputs to all the decoders since during inference there is no input for the action token corresponding to the current timestep (this will be a_t, exactly what we aim to predict). The action tokens in the past are used in the attention layers, though. The alternative is to add an extra prompt token for the current actions. We stick to the standard approach of using the state tokens for inputs to the decoders as in DT/BeT.
> 5. The content of the CoT tokens is the learnable prompt embeddings. Each CoT token has its embeddings (not shared since we do not use position embedding for the CoT tokens). They are fixed during inference.
> 6. The T CoT tokens correspond to the T timesteps in the context length of the Transformer. Each token is learned to predict the same CoT content (next and last subgoal). There is no alignment issue between subgoals and CoT tokens in this setup. Because we couple action prediction and CoT prediction, we need T branches for T action offset predictions in the first place. We find the alternative producing less generalizable policies.
>
> **BeT should have been explained more**
>
> We’ve added a brief intro to the BeT architecture in the appendix.
>
> **The third ablation study needs a diagram to illustrate the different variants of the architecture**
>
> We’ve added a diagram in the appendix to illustrate the differences.
>
> **Curious to see what the effect of setting the component of the auxiliary loss term to 0, but keeping learnable prompt tokens (maintain the same capacity as the CoTPC architecture)**
>
> We perform an ablation study on Peg Insertion and Push Chair with state observation (similar to our other ablation studies) where we simply add additional prompt tokens to the BeT baseline (no auxiliary loss and thus no CoT decoder). This variant (named BeT+prompt) has the same capacity as CoTPC yet performs very similarly to the BeT baseline (shown below). We report unseen SR for Push Chair and 0-shot SR for Peg Insertion (as in our other ablation studies).
>
> |    | BeT | BeT+prompt | CoTPC |
> | -------- | ------- | ------- | ------- |
> | Peg Insertion | 42.5 | 43.0 | 59.3 |
> | Push Chair | 33.4 | 32.7 | 41.0 |
>
> **Limitations of the approach**
>
> We will add a paragraph to discuss the limitation of action-based subskill discovery. Specifically, the assumption that similar actions that are also temporarily close should be grouped into the same subskill can be violated with complex manipulations with many micro-adjustments (e.g., standing and balancing on a rope). In this case, extra force feedback might be required to distinguish subskills that are only subtly different.

---

> ### Author Response · Authors · 2023-11-23
> **Additional Author Response**
>
> In the Maze task, since the state observation is essentially a vector of the position of the agent as well as the moving blocks, we find that the CoT tokens and the CoT decoder learn to predict the positions of the moving blocks relatively precisely from which the agent will reach them. Since it is a relatively simple task with very clear subgoals, the illustration provides less information than that of the ManiSkill2 tasks. We will add it to the Appendix if required.

---

### Meta-Review · Area_Chair_7FNP · 2023-12-10

**Metareview:**

This paper presents a method for hierarchical planning from the demonstrations using a Transformer-based architecture. The demonstrations are decomposed into sequences of observations and are used for unsupervised discovery of subgoals. Experiments on the Moving Maze, Franka Kitchen and ManiSkill2 environments are used to evaluate the method.

The reviewers are split evenly (two 5s, two 6s).

Strengths of the paper include good motivation, well-written, good experimental set up and in general addressing an important and complex problem.

Weaknesses of the paper include limited novelty, limited experimentation and several points raised by reviewer QpAT on writing and comparison with existing methods. Looking into the discussion it is clear that the paper can be further improved with appropriate discussions as suggested by QpAT.

**Justification For Why Not Higher Score:**

The critical reviewer makes a compelling argument that the paper should be improved. The two positive reviews while positive does not substantially elaborate on positives of the paper.

**Justification For Why Not Lower Score:**

N/A

---

### Decision · Program_Chairs · 2024-01-16

Reject